# Learning spatiotemporal trajectories from manifold-valued longitudinal data

**Jean-Baptiste Schiratti**[2,1]**, Stéphanie Allassonnière**[2]**, Olivier Colliot**[1]**, Stanley Durrleman**[1]
[1] ARAMIS Lab, INRIA Paris, Inserm U1127, CNRS UMR 7225, Sorbonne Universités,
UPMC Univ Paris 06 UMR S 1127, Institut du Cerveau et de la Moelle épinière,
ICM, F-75013, Paris, France
[2]CMAP, Ecole Polytechnique, Palaiseau, France
jean-baptiste.schiratti@cmap.polytechnique.fr,
stephanie.allassonniere@polytechnique.edu,
olivier.colliot@upmc.fr,stanley.durrleman@inria.fr

## Abstract

We propose a Bayesian mixed-effects model to learn typical scenarios of changes from longitudinal manifold-valued data, namely repeated measurements of the same objects or individuals at several points in time. The model allows to estimate a group-average trajectory in the space of measurements. Random variations of this trajectory result from spatiotemporal transformations, which allow changes in the direction of the trajectory and in the pace at which trajectories are followed. The use of the tools of Riemannian geometry allows to derive a generic algorithm for any kind of data with smooth constraints, which lie therefore on a Riemannian manifold. Stochastic approximations of the Expectation-Maximization algorithm is used to estimate the model parameters in this highly non-linear setting. The method is used to estimate a data-driven model of the progressive impairments of cognitive functions during the onset of Alzheimer's disease. Experimental results show that the model correctly put into correspondence the age at which each individual was diagnosed with the disease, thus validating the fact that it effectively estimated a normative scenario of disease progression. Random effects provide unique insights into the variations in the ordering and timing of the succession of cognitive impairments across different individuals.

## 1 Introduction

Age-related brain diseases, such as Parkinson's or Alzheimer's disease (AD) are complex diseases with multiple effects on the metabolism, structure and function of the brain. Models of disease progression showing the sequence and timing of these effects during the course of the disease remain largely hypothetical [3, 13]. Large databases have been collected recently in the hope to give experimental evidence of the patterns of disease progression based on the estimation of data-driven models. These databases are longitudinal, in the sense that they contain repeated measurements of several subjects at multiple time-points, but which do not necessarily correspond across subjects.

Learning models of disease progression from such databases raises great methodological challenges. The main difficulty lies in the fact that the age of a given individual gives no information about the stage of disease progression of this individual. The onset of clinical symptoms of AD may vary from forty and eighty years of age, and the duration of the disease from few years to decades. Moreover, the onset of the disease does not correspond with the onset of the symptoms: according to recent studies, symptoms are likely to be preceded by a silent phase of the disease, for which little is known. As a consequence, statistical models based on the regression of measurements with age are inadequate to model disease progression.

The set of the measurements of a given individual at a specific time-point belongs to a high-dimensional space. Building a model of disease progression amounts to estimating continuous subject-specific trajectories in this space and average those trajectories among a group of individuals. Trajectories need to be registered in *space*, to account for the fact that individuals follow different trajectories, and in *time*, to account for the fact that individuals, even if they follow the same trajectory, may be at a different position on this trajectory at the same age.

The framework of mixed-effects models seems to be well suited to deal with this hierarchical problem. Mixed-effects models for longitudinal measurements were introduced in the seminal paper of Laird and Ware [15] and have been widely developed since then (see [6], [16] for instance). However, this kind of models suffers from two main drawbacks regarding our problem. These models are built on the estimation of the distribution of the measurements at a given time point. In many situations, this reference time is given by the experimental set up: date at which treatment begins, date of seeding in studies of plant growth, etc. In studies of ageing, using these models would require to register the data of each individual to a common stage of disease progression before being compared. Unfortunately, this stage is unknown and such a temporal registration is actually what we wish to estimate. Another limitation of usual mixed-effects models is that they are defined for data lying in Euclidean spaces. However, measurements with smooth constraints usually cannot be summed up or scaled, such as normalized scores of neurospychological tests, positive definite symmetric matrices, shapes encoded as images or meshes. These data are naturally modeled as points on Riemannian manifolds. Although the development of statistical models for manifold-valued data is a blooming topic, the construction of statistical models for longitudinal data on a manifold remains an open problem.

The concept of "time-warp" was introduced in [8] to allow for temporal registration of trajectories of shape changes. Nevertheless, the combination of the time-warps with the intrinsic variability of shapes across individuals is done at the expense of a simplifying approximation: the variance of shapes does not depend on time whereas it should adapt with the average scenario of shape changes. Moreover, the estimation of the parameters of the statistical model is made by minimizing a sum of squares which results from an uncontrolled likelihood approximation. In [18], time-warps are used to define a metric between curves that are invariant under time reparameterization. This invariance, by definition, prevents the estimation of correspondences across trajectories, and therefore the estimation of distribution of trajectories in the spatiotemporal domain. In [17], the authors proposed a model for longitudinal image data but the model is not built on the inference of a statistical model and does not include a time reparametrization of the estimated trajectories.

In this paper, we propose a generic statistical framework for the definition and estimation of mixed-effects models for longitudinal manifold-valued data. Using the tools of geometry allows us to derive a method that makes little assumptions about the data and problem to deal with. Modeling choices boil down to the definition of the metric on the manifold. This geometrical modeling also allows us to introduce the concept of parallel curves on a manifold, which is key to uniquely decompose differences seen in the data in a spatial and a temporal component. Because of the non-linearity of the model, the estimation of the parameters should be based on an adequate maximization of the observed likelihood. To address this issue, we propose to use a stochastic version of the Expectation-Maximization algorithm [5], namely the MCMC SAEM [2], for which theoretical results regarding the convergence have been proved in [4], [2].

Experimental results on neuropsychological tests scores and estimates of scenarios of AD progression are given in section 4.

## 2 Spatiotemporal mixed-effects model for manifold-valued data

### 2.1 Riemannian geometry setting

The observed data consists in repeated multivariate measurements of $p$ individuals. For a given individual, the measurements are obtained at time points $t_{i,1} < \ldots < t_{i,n_i}$. The $j$-th measurement of the $i$-th individual is denoted by $\mathbf{y}_{i,j}$. We assume that each observation $\mathbf{y}_{i,j}$ is a point on a $N$-dimensional Riemannian manifold $\mathbb{M}$ embedded in $\mathbb{R}^P$ (with $P \geq N$) and equipped with a Riemannian metric $g^{\mathbb{M}}$. We denote $\nabla^{\mathbb{M}}$ the covariant derivative. We assume that the manifold is geodesically complete, meaning that geodesics are defined for all time.

We recall that a geodesic is a curve drawn on the manifold $\gamma : \mathbb{R} \to \mathbb{M}$, which has no acceleration: $\nabla_{\dot{\gamma}}^{\mathbb{M}} \dot{\gamma} = 0$. For a point $\mathbf{p} \in \mathbb{M}$ and a vector $\mathbf{v} \in \mathrm{T}_{\mathbf{p}}\mathbb{M}$, the mapping $\mathrm{Exp}_{\mathbf{p}}^{\mathbb{M}}(\mathbf{v})$ denotes the Riemannian exponential, namely the point that is reached at time 1 by the geodesic starting at $\mathbf{p}$ with velocity $\mathbf{v}$. The parallel transport of a vector $X_0 \in \mathrm{T}_{\gamma(t_0)}\mathbb{M}$ in the tangent space at point $\gamma(t_0)$ on a curve $\gamma$ is a time-indexed family of vectors $X(t) \in \mathrm{T}_{\gamma(t)}\mathbb{M}$ which satisfies $\nabla_{\dot{\gamma}(t)}^{\mathbb{M}} X(t) = 0$ and $X(t_0) = X_0$. We denote $\mathrm{P}_{\gamma, t_0, t}(X_0)$ the isometry that maps $X_0$ to $X(t)$.

In order to describe our model, we need to introduce the notion of "parallel curves" on the manifold:

**Definition 1.** Let $\gamma$ be a curve on $\mathbb{M}$ defined for all time, a time-point $t_0 \in \mathbb{R}$ and a vector $\mathbf{w} \in \mathrm{T}_{\gamma(t_0)}\mathbb{M}$, $\mathbf{w} \neq 0$. One defines the curve $s \to \eta^{\mathbf{w}}(\gamma, s)$, called parallel to the curve $\gamma$, as:

$$\eta^{\mathbf{w}}(\gamma, s) = \mathrm{Exp}_{\gamma(s)}^{\mathbb{M}}\big(\mathrm{P}_{\gamma, t_0, s}(\mathbf{w})\big), \ s \in \mathbb{R}.$$

The idea is illustrated in Fig. 1. One uses the parallel transport to move the vector $\mathbf{w}$ from $\gamma(t_0)$ to $\gamma(s)$ along $\gamma$. At the point $\gamma(s)$, a new point on $\mathbb{M}$ is obtained by taking the Riemannian exponential of $\mathrm{P}_{\gamma, t_0, s}(\mathbf{w})$. This new point is denoted by $\eta^{\mathbf{w}}(\gamma, s)$. As $s$ varies, one describes a curve $\eta^{\mathbf{w}}(\gamma, \cdot)$ on $\mathbb{M}$, which can be understood as a "parallel" to the curve $\gamma$. It should be pointed out that, even if $\gamma$ is a geodesic, $\eta^{\mathbf{w}}(\gamma, \cdot)$ is, in general, not a geodesic of $\mathbb{M}$. In the Euclidean case (i.e. a flat manifold), the curve $\eta^{\mathbf{w}}(\gamma, \cdot)$ is the translation of the curve $\gamma$: $\eta^{\mathbf{w}}(\gamma, s) = \gamma(s) + \mathbf{w}$.

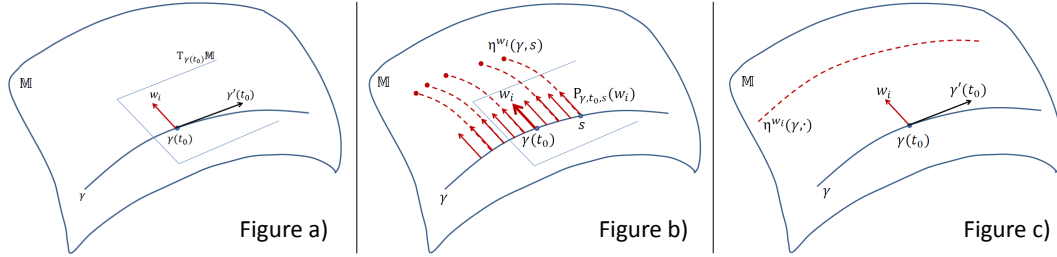

| Figure a) | Figure b) | Figure c) |

Figure 1: Model description on a schematic manifold. Figure a) (left) : a non-zero vector $\mathbf{w}_i$ is choosen in $\mathrm{T}_{\gamma(t_0)}\mathbb{M}$. Figure b) (middle) : the tangent vector $\mathbf{w}_i$ is transported along the geodesic $\gamma$ and a point $\eta^{\mathbf{w}_i}(\gamma, s)$ is constructed at time $s$ by use of the Riemannian exponential. Figure c) (right) : The curve $\eta^{\mathbf{w}_i}(\gamma, \cdot)$ is the parallel resulting from the construction.

## 2.2 Generic spatiotemporal model for longitudinal data

Our model is built in a hierarchical manner: data points are seen as samples along individual trajectories, and these trajectories derive from a group-average trajectory. The model writes $\mathbf{y}_{i,j} = \eta^{\mathbf{w}_i}(\gamma, \psi_i(t_{i,j})) + \varepsilon_{i,j}$, where we assume the group-average trajectory to be a geodesic, denoted $\gamma$ from now on. Individual trajectories derive from the group average by spatiotemporal transformations. They are defined as a time re-parameterization of a trajectory that is parallel to the group-average: $t \to \eta^{\mathbf{w}_i}(\gamma, \psi_i(t))$. For the i*th* individual, $\mathbf{w}_i$ denotes a non-zero tangent vector in $\mathrm{T}_{\gamma(t_0)}\mathbb{M}$, for some specific time point $t_0$ that needs to be estimated, and which is orthogonal to the tangent vector $\dot{\gamma}(t_0)$ for the inner product given by the metric ($\langle \cdot, \cdot \rangle_{\gamma(t_0)} = g_{\gamma(t_0)}^{\mathbb{M}}$). The time-warp function $\psi_i$ is defined as: $\psi_i(t) = \alpha_i(t - t_0 - \tau_i) + t_0$. The parameter $\alpha_i$ is an *acceleration factor* which encodes whether the $i$-th individual is progressing faster or slower than the average, $\tau_i$ is a *time-shift* which characterizes the advance or delay of the *ith* individual with respect to the average and $\mathbf{w}_i$ is a *space-shift* which encodes the variability in the measurements across individuals at the same stage of progression.

The *normal tubular neighborhood theorem* ([11]) ensures that parallel shifting defines a spatiotemporal coordinate system as long as the vectors $w_i$ are choosen orthogonal and sufficently small. The orthogonality condition on the tangent vectors $\mathbf{w}_i$ is necessary to ensure the identifiability of the model. Indeed, if a vector $\mathbf{w}_i$ was not choosen orthogonal, its orthogonal projection would play the same role as the acceleration factor.The spatial and temporal transformations commute, in the sense that one may re-parameterize the average trajectory before building the parallel curve, or vice versa. Mathematically, this writes $\eta^{\mathbf{w}_i}(\gamma \circ \psi_i, s) = \eta^{\mathbf{w}_i}(\gamma, \psi_i(s))$. This relation also explains the particular form of the affine time-warp $\psi_i$. The geodesic $\gamma$ is characterized by the fact that it passes

at time-point $t_0$ by point $\mathbf{p}_0 = \boldsymbol{\gamma}(t_0)$ with velocity $\mathbf{v}_0 = \dot{\gamma}(t_0)$. Then, $\boldsymbol{\gamma} \circ \psi_i$ is the same trajectory, except that it passes by point $\mathbf{p}_0$ at time $t_0 + \tau_i$ with velocity $\alpha_i \mathbf{v}_0$.

The fixed effects of the model are the parameters of the average geodesic: the point $\mathbf{p}_0$ on the manifold, the time-point $t_0$ and the velocity $\mathbf{v}_0$. The random effects are the acceleration factors $\alpha_i$, time-shifts $\tau_i$ and space-shifts $\mathbf{w}_i$. The first two random effects are scalars. One assumes the acceleration factors to follow a log-normal distribution (they need to be positive in order not to reverse time), and time-shifts to follow a zero-mean Gaussian distribution. Space-shifts are vectors of dimension $N-1$ in the hyperplane $\dot{\boldsymbol{\gamma}}(t_0)^\perp$ in $\mathrm{T}_{\boldsymbol{\gamma}(t_0)}\mathbb{M}$. In the spirit of independent component analysis [12], we assume that $\mathbf{w}_i$'s result from the superposition of $N_s < N$ statistically independent components. This writes $\mathbf{w}_i = \mathbf{A}\mathbf{s}_i$ where $\mathbf{A}$ is a $N \times N_s$ matrix of rank $N_s$, $\mathbf{s}_i$ a vector of $N_s$ independent sources following a heavy tail Laplace distribution with fixed parameter, and each column $\mathbf{c}_j(\mathbf{A})$ $(1 \leq j \leq N_s)$ of $\mathbf{A}$ satisfies the orthogonality condition $\langle \mathbf{c}_j(\mathbf{A}), \dot{\boldsymbol{\gamma}}(t_0) \rangle_{\boldsymbol{\gamma}(t_0)} = 0$.

For the dataset $(t_{i,j}, \mathbf{y}_{i,j})$ $(1 \leq i \leq p, \ 1 \leq j \leq n_i)$, the model may be summarized as:

$$\mathbf{y}_{i,j} = \eta^{\mathbf{w}_i}(\boldsymbol{\gamma}, \psi_i(t_{i,j})) + \boldsymbol{\varepsilon}_{i,j}. \tag{1}$$

with $\psi_i(t) = \alpha_i(t - t_0 - \tau_i) + t_0$, $\alpha_i = \exp(\xi_i)$, $\mathbf{w}_i = \mathbf{A}\mathbf{s}_i$ and

$$\xi_i \overset{\text{i.i.d.}}{\sim} \mathcal{N}(0, \sigma_\eta^2), \ \tau_i \overset{\text{i.i.d.}}{\sim} \mathcal{N}(0, \sigma_\tau^2), \ \boldsymbol{\varepsilon}_{i,j} \overset{\text{i.i.d.}}{\sim} \mathcal{N}(0, \sigma^2 \mathrm{I}_N), \ s_{i,l} \overset{\text{i.i.d.}}{\sim} \text{Laplace}(1/2).$$

Eventually, the parameters of the model one needs to estimate are the fixed effects and the variance of the random effects, namely $\boldsymbol{\theta} = (\mathbf{p}_0, t_0, \mathbf{v}_0, \sigma_\xi, \sigma_\tau, \sigma, \text{vec}(\mathbf{A}))$.

## 2.3 Propagation model in a product manifold

We wish to use these developments to study the temporal progression of a family of biomarkers. We assume that each component of $\mathbf{y}_{i,j}$ is a scalar measurement of a given biomarker and belongs to a geodesically complete one-dimensional manifold $(M, g)$. Therefore, each measurement $\mathbf{y}_{i,j}$ is a point in the product manifold $\mathbb{M} = M^N$, which we assume to be equipped with the Riemannian product metric $g^{\mathbb{M}} = g + \ldots + g$. We denote $\boldsymbol{\gamma}_0$ the geodesic of the one-dimensional manifold $M$ which goes through the point $p_0 \in M$ at time $t_0$ with velocity $v_0 \in \mathrm{T}_{p_0}M$. In order to determine relative progression of the biomarkers among themselves, we consider a parametric family of geodesics of $\mathbb{M}$: $\boldsymbol{\gamma}_{\boldsymbol{\delta}}(t) = (\gamma_0(t), \gamma_0(t+\delta_1), \ldots, \gamma_0(t+\delta_{N-1}))$. We assume here that all biomarkers have on average the same dynamics but shifted in time. This hypothesis allows to model a temporal succession of effects during the course of the disease. The relative timing in biomarker changes is measured by the vector $\boldsymbol{\delta} = (0, \delta_1, \ldots, \delta_{N-1})$, which becomes a fixed effect of the model.

In this setting, a curve that is parallel to a geodesic $\boldsymbol{\gamma}$ is given by the following lemma :

**Lemma 1.** Let $\boldsymbol{\gamma}$ be a geodesic of the product manifold $\mathbb{M} = M^N$ and let $t_0 \in \mathbb{R}$. If $\eta^{\mathbf{w}}(\boldsymbol{\gamma}, \cdot)$ denotes a parallel to the geodesic $\boldsymbol{\gamma}$ with $\mathbf{w} = (w_1, \ldots, w_N) \in \mathrm{T}_{\boldsymbol{\gamma}(t_0)}\mathbb{M}$ and $\boldsymbol{\gamma}(t) = (\gamma_1(t), \ldots, \gamma_N(t))$, we have $\eta^{\mathbf{w}}(\boldsymbol{\gamma}, s) = (\gamma_1(\frac{w_1}{\dot{\gamma}(t_0)} + s), \ldots, \gamma_N(\frac{w_N}{\dot{\gamma}(t_0)} + s))$, $s \in \mathbb{R}$.

As a consequence, a parallel to the average trajectory $\boldsymbol{\gamma}_{\boldsymbol{\delta}}$ has the same form as the geodesic but with randomly perturbed delays. The model (1) writes : for all $k \in \{1, \ldots, N\}$,

$$y_{i,j,k} = \gamma_0 \Big( \frac{w_{k,i}}{\dot{\gamma}_0(t_0 + \delta_{k-1})} + \alpha_i(t_{i,j} - t_0 - \tau_i) + t_0 + \delta_{k-1} \Big) + \varepsilon_{i,j,k}. \tag{2}$$

where $w_{k,i}$ denotes the $k$-th component of the space-shift $w_i$ and $y_{i,j,k}$, the measurement of the $k$-th biomarker, at the $j$-th time point, for the $i$-th individual.

## 2.4 Multivariate logistic curves model

The propagation model given in (2) is now described for *normalized* biomarkers, such as scores of neuropsychological tests. In this case, we assume the manifold to be $M = ]0, 1[$ and equipped with the Riemannian metric $g$ given by : for $p \in ]0, 1[$, $(u, v) \in \mathrm{T}_pM \times \mathrm{T}_pM$, $g_p(u, v) = u\mathrm{G}(p)v$ with $\mathrm{G}(p) = 1/(p^2(1-p)^2)$. The geodesics given by this metric in the one-dimensional Riemannian manifold $M$ are logistic curves of the form : $\gamma_0(t) = \big(1 + (\frac{1}{p_0} - 1)\exp\big(-\frac{v_0}{p_0(1-p_0)}(t - t_0)\big)\big)^{-1}$ and leads to the *multivariate logistic curves model* in $\mathbb{M}$. We can notice the quite unusual paramaterization of the logistic curve. This parametrization naturally arise because $\gamma_0$ satisfies : $\gamma_0(t_0) = p_0$ and $\dot{\gamma}_0(t_0) = v_0$. In this case, the model (1) writes:

$$y_{i,j,k} = \left(1 + \left(\frac{1}{p_0} - 1\right) \exp\left(-\frac{v_0\alpha_i(t_{i,j} - t_0 - \tau_i) + v_0\delta_k + v_0\frac{(\mathbf{As}_i)_k}{\dot{\gamma_0}(t_0+\delta_k)}}{p_0(1-p_0)}\right)\right)^{-1} + \varepsilon_{i,j,k}, \quad (3)$$

where $(\mathbf{As}_i)_k$ denotes the $k$-th component of the vector $\mathbf{As}_i$. Note that (3) is not equivalent to a linear model on the logit of the observations. The logit transform corresponds to the Riemannian logarithm at $p_0 = 0.5$. In our framework, $p_0$ is not fixed, but estimated as a parameter of our model. Even with a fixed $p_0 = 0.5$, the model is still *non-linear* due to the multiplication between random-effects $\alpha_i$ and $\tau_i$, and therefore does not boil down to the usual linear model [15].

## 3   Parameters estimation

In this section, we explain how to use a stochastic version of the Expectation-Maximization (EM) algorithm [5] to produce estimates of the parameters $\boldsymbol{\theta} = (\mathbf{p}_0, t_0, \mathbf{v}_0, \boldsymbol{\delta}, \sigma_\xi, \sigma_\tau, \sigma, \text{vec}(\mathbf{A}))$ of the model. The algorithm detailed in this section is essentially the same as in [2]. Its scope of application is not limited to statistical models on product manifolds and the MCMC-SAEM algorithm can actually be used for the inference of a very large family of statistical models.

The random effects $\mathbf{z} = (\xi_i, \tau_i, s_{j,i})$ $(1 \leq i \leq p$ and $1 \leq j \leq N_s)$ are considered as hidden variables. With the observed data $\mathbf{y} = (y_{i,j,k})_{i,j,k}$, $(\mathbf{y}, \mathbf{z})$ form the complete data of the model. In this context, the Expectation-Maximization (EM) algorithm proposed in [5] is very efficient to compute the maximum likelihood estimate of $\boldsymbol{\theta}$. Due to the nonlinearity and complexity of the model, the E step is intractable. As a consequence, we considered a stochastic version of the EM algorithm, namely the Monte-Carlo Markov Chain Stochastic Approximation Expectation-Maximization (MCMC-SAEM) algorithm [2], based on [4]. This algorithm is an EM-like algorithm which alternates between three steps: simulation, stochastic approximation and maximization. If $\boldsymbol{\theta}^{(t)}$ denotes the current parameter estimates of the algorithm, in the simulation step, a sample $\mathbf{z}^{(t)}$ of the missing data is obtained from the transition kernel of an ergodic Markov chain whose stationary distribution is the conditional distribution of the missing data $\mathbf{z}$ knowing $\mathbf{y}$ and $\boldsymbol{\theta}^{(t)}$, denoted by $q(\mathbf{z} \,|\, \mathbf{y}, \boldsymbol{\theta}^{(t)})$. This simulation step is achieved using Hasting-Metropolis within Gibbs sampler. Note that the high complexity of our model prevents us from resorting to sampling methods as in [10] as they would require heavy computations, such as the Fisher information matrix. The stochastic approximation step consists in a stochastic approximation on the complete log-likelihood $\log q(\mathbf{y}, \mathbf{z} \,|\, \boldsymbol{\theta})$ summarized as follows : $Q_t(\boldsymbol{\theta}) = Q_{t-1}(\boldsymbol{\theta}) + \varepsilon_t \left[\log q(\mathbf{y}, \mathbf{z} \,|\, \boldsymbol{\theta}) - Q_{t-1}(\boldsymbol{\theta})\right]$, where $(\varepsilon_t)_t$ is a decreasing sequence of positive step-sizes in $]0, 1]$ which satisfies $\sum_t \varepsilon_t = +\infty$ and $\sum_t \varepsilon_t^2 < +\infty$. Finally, the parameter estimates are updated in the maximization step according to: $\boldsymbol{\theta}^{(t+1)} = \text{argmax}_{\boldsymbol{\theta} \in \Theta} Q_t(\boldsymbol{\theta})$.

The theoretical convergence of the MCMC SAEM algorithm is proved only if the model belong to the curved exponential family. Or equivalently, if the complete log-likelihood of the model may be written : $\log q(\mathbf{y}, \mathbf{z} \,|\, \boldsymbol{\theta}) = -\phi(\boldsymbol{\theta}) + S(\mathbf{y}, \mathbf{z})^\top \psi(\boldsymbol{\theta})$, where $S(\mathbf{y}, \mathbf{z})$ is a sufficent statistic of the model. In this case, the stochastic approximation on the complete log-likelihood can be replaced with a stochastic approximation on the sufficent statistics of the model. Note that the *multivariate logistic curves model* does not belong to the curved exponential family. A usual workaround consists in regarding the parameters of the model as realizations of independents Gaussian random variables ([14]) : $\boldsymbol{\theta} \sim \mathcal{N}(\overline{\boldsymbol{\theta}}, \mathbf{D})$ where $\mathbf{D}$ is a diagonal matrix with very small diagonal entries and the estimation now targets $\overline{\boldsymbol{\theta}}$. This yields: $p_0 \sim \mathcal{N}(\overline{p_0}, \sigma_{p_0}^2)$, $t_0 \sim \mathcal{N}(\overline{t_0}, \sigma_{t_0}^2)$, $v_0 \sim \mathcal{N}(\overline{v_0}, \sigma_{v_0}^2)$ and, for all $k$, $\delta_k \sim \mathcal{N}(\overline{\delta}_k, \sigma_\delta^2)$. To ensure the orthogonality condition on the columns of $\mathbf{A}$, we assumed that $\mathbf{A}$ follows a normal distribution on the space $\boldsymbol{\Sigma} = \{\mathbf{A} = (\mathbf{c}_1(\mathbf{A}), \ldots, \mathbf{c}_{N_s}(\mathbf{A})) \in \left(\mathrm{T}_{\gamma_\delta(t_0)}\mathbb{M}\right)^{N_s} ; \forall j, \langle \mathbf{c}_j(\mathbf{A}), \dot{\gamma}_\delta(t_0)\rangle_{\gamma_\delta(t_0)} = 0\}$. Equivalently, we assume that the matrix $\mathbf{A}$ writes : $\mathbf{A} = \sum_{k=1}^{(N-1)N_s} \beta_k \mathcal{B}_k$ where, for all $k$, $\beta_k \overset{\text{i.i.d.}}{\sim} \mathcal{N}(\overline{\beta}_k, \sigma_\beta^2)$ and $(\mathcal{B}_1, \ldots, \mathcal{B}_{(N-1)N_s})$ is an orthonormal basis of $\boldsymbol{\Sigma}$ obtained by application of the Gram-Schmidt process to a basis of $\boldsymbol{\Sigma}$. The random variables $\beta_1, \ldots, \beta_{(N-1)N_s}$ are considered as new hidden variables of the model. The parameters of the model are $\boldsymbol{\theta} = (\overline{p_0}, \overline{t_0}, \overline{v_0}, (\overline{\delta}_k)_{1 \leq k \leq N-1}, (\overline{\beta}_k)_{1 \leq k \leq (N-1)N_s}, \sigma_\xi, \sigma_\tau, \sigma)$ whereas the hidden variables of the model are $\mathbf{z} = (p_0, t_0, v_0, (\delta_k)_{1 \leq k \leq N-1}, (\beta_k)_{1 \leq k \leq (N-1)N_s}, (\xi_i)_{1 \leq i \leq p}, (\tau_i)_{1 \leq i \leq p}, (s_{j,i})_{1 \leq j \leq N_s, 1 \leq i \leq p})$. The algorithm (1) given below summarizes the SAEM algorithm for this model.

The MCMC-SAEM algorithm 1 was tested on synthetic data generated according to (3). The MCMC-SAEM allowed to recover the parameters used to generate the synthetic dataset.

**Algorithm 1** Overview of the MCMC SAEM algorithm for the *multivariate logistic curves model*.

If $\mathbf{z}^{(k)} = \big(p_0^{(k)}, t_0^{(k)}, \ldots, (s_{j,i}^{(k)})\big)$ denotes the vector of hidden variables obtained in the simulation step of the $k$-th iteration of the MCMC SAEM, let $\mathbf{f}_{i,j} = [f_{i,j,l}] \in \mathbb{R}^N$ and $f_{i,j,l}$ be the $l$-th component of $\eta_{w_i^{(k)}}\big((\boldsymbol{\gamma_\delta})^{(k)}, \exp(\xi_i^{(k)})(t_{i,j} - t_0^{(k)} - \tau_i^{(k)}) + t_0^{(k)}\big)$ and $w_i^{(k)} = \sum_{l=1}^{(N-1)N_s} \beta_l^{(k)} \mathcal{B}_l$.

> Initialization :
> $\boldsymbol{\theta} \leftarrow \boldsymbol{\theta}^{(0)}$ ; $\mathbf{z}^{(0)} \leftarrow$ random ; $\mathbf{S} \leftarrow 0$ ; $(\varepsilon_k)_{k \geq 0}$.
> **repeat**
>> Simulation step : $\mathbf{z}^{(k)} = \big(p_0^{(k)}, t_0^{(k)}, \ldots, (s_{j,i}^{(k)})_{j,i}\big) \leftarrow$ Gibbs Sampler$(\mathbf{z}^{(k-1)}, \mathbf{y}, \boldsymbol{\theta}^{(k-1)})$
>> Compute the suffcient statistics : $\mathbf{S}_1^{(k)} \leftarrow \big[\mathbf{y}_{i,j}^\top \mathbf{f}_{i,j}\big]_{i,j} \in \mathbb{R}^K$ ; $\mathbf{S}_2^{(k)} \leftarrow \big[\|\mathbf{f}_{i,j}\|^2\big]_{i,j} \in \mathbb{R}^K$
>> with $(1 \leq i \leq p ; 1 \leq j \leq n_i)$ and $K = \sum_{i=1}^p n_i$ ; $\mathbf{S}_3^{(k)} = \big[(\xi_i^{(k)})^2\big]_i \in \mathbb{R}^p$ ; $\mathbf{S}_4^{(k)} \leftarrow$
>> $\big[(\tau_i^{(k)})^2\big]_i \in \mathbb{R}^p$ ; $\mathbf{S}_5^{(k)} \leftarrow p_0^{(k)}$ ; $\mathbf{S}_6^{(k)} \leftarrow t_0^{(k)}$ ; $\mathbf{S}_7^{(k)} \leftarrow v_0^{(k)}$ ; $\mathbf{S}_8^{(k)} \leftarrow \big[\delta_j^{(k)}\big]_j \in \mathbb{R}^{N-1}$ ;
>> $\mathbf{S}_9^{(k)} \leftarrow \big[\beta_j^{(k)}\big]_j \in \mathbb{R}^{(N-1)N_s}$.
>> Stochastic approximation step : $\mathbf{S}_j^{(k+1)} \leftarrow \mathbf{S}_j^{(k)} + \varepsilon_k(\mathbf{S}(\mathbf{y}, \mathbf{z}^{(k)}) - \mathbf{S}_j^{(k)})$ for $j \in \{1, \ldots, 9\}$.
>> Maximization step : $\overline{p_0}^{(k+1)} \leftarrow \mathbf{S}_5^{(k)}$ ; $\overline{t_0}^{(k+1)} \leftarrow \mathbf{S}_6^{(k)}$ ; $\overline{v_0}^{(k+1)} \leftarrow \mathbf{S}_7^{(k)}$ ; $\overline{\delta}_j^{(k+1)} \leftarrow (\mathbf{S}_8^{(k)})_j$
>> for all $1 \leq j \leq N-1$ ; $\overline{\beta}_j^{(k+1)} \leftarrow (\mathbf{S}_9^{(k)})_j$ for all $1 \leq j \leq (N-1)N_s$ ; $\sigma_\xi^{(k+1)} \leftarrow \frac{1}{p}(\mathbf{S}_3^{(k)})^\top \mathbb{1}_p$
>> ; $\sigma_\tau^{(k+1)} \leftarrow \frac{1}{p}(\mathbf{S}_4^{(k)})^\top \mathbb{1}_p$ ; $\sigma^{(k+1)} \leftarrow \frac{1}{\sqrt{NK}}\big(\sum_{i,j,k} y_{i,j,k}^2 - 2(\mathbf{S}_1^{(k)})^\top \mathbb{1}_K + (\mathbf{S}_2^{(k)})^\top \mathbb{1}_K\big)^{1/2}$.
> **until** convergence.
> **return** $\theta$.

## 4 Experiments

### 4.1 Data

We use the neuropsychological assessment test "ADAS-Cog 13" from the ADNI1, ADNIGO or ADNI2 cohorts of the Alzheimer's Disease Neuroimaging Initiative (ADNI) [1]. The "ADAS-Cog 13" consists of 13 questions, which allow to test the impairment of several cognitive functions. For the purpose of our analysis, these items are grouped into four categories: memory (5 items), language (5 items), praxis (2 items) and concentration (1 item). Scores within each category are added and normalized by the maximum possible score. Consequently, each data point consists in four normalized scores, which can be seen as a point on the manifold $\mathbb{M} = ]0, 1[^4$.

We included 248 individuals in the study, who were diagnosed with mild cognitive impairment (MCI) at their first visit and whose diagnosis changed to AD before their last visit. There is an average of 6 visits per subjects (min: 3, max: 11), with an average duration of 6 or 12 months between consecutive visits. The *multivariate logistic curves* model was used to analyze this longitudinal data.

### 4.2 Experimental results

The model was applied with $N_s = 1, 2$ or $3$ independent sources. In each experiment, the MCMC SAEM was run five times with different initial parameter values. The experiment which returned the smallest residual variance $\sigma^2$ was kept. The maximum number of iterations was arbitrarily set to 5000 and the number of burn-in iterations was set to 3000 iterations. The limit of 5000 iterations is enough to observe the convergence of the sequences of parameters estimates. As a result, two and three sources allowed to decrease the residual variance better than one source ($\sigma^2 = 0.012$ for one source, $\sigma^2 = 0.08$ for two sources and $\sigma^2 = 0.084$ for three sources). The residual variance $\sigma^2 = 0.012$ (resp. $\sigma^2 = 0.08$, $\sigma^2 = 0.084$) mean that the model allowed to explain 79% (resp. 84%, 85%) of the total variance. We implemented our algorithm in MATLAB without any particular optimization scheme. The 5000 iterations require approximately one day.

The number of parameters to be estimated is equal to $9 + 3N_s$. Therefore, the number of sources do not dramatically impact the runtime. Simulation is the most computationally expensive part of

our algorithm. For each run of the Hasting-Metropolis algorithm, the proposal distribution is the prior distribution. As a consequence, the acceptation ratio simplifies [2] and one computation of the acceptation ratio requires two computations of the likelihood of the observations, conditionally on different vectors of latent variables and the vector of current parameters estimates. The runtime could be improved by parallelizing the sampling per individuals.

For a matter of clarity and because the results obtained with three sources were similar to the results with two sources, we report here the experimental results obtained with two independent sources. The average model of disease progression $\gamma_\delta$ is plotted in Fig. 2. The estimated fixed effects are $p_0 = 0.3$, $t_0 = 72$ years, $v_0 = 0.04$ unit per year, and $\boldsymbol{\delta} = [0; -15; -13; -5]$ years. This means that, on average, the memory score (first coordinate) reaches the value $p_0 = 0.3$ at $t_0 = 72$ years, followed by concentration which reaches the same value at $t_0 + 5 = 77$ years, and then by praxis and language at age 85 and 87 years respectively.

Random effects show the variability of this average trajectory within the studied population. The standard deviation of the time-shift equals $\sigma_\tau = 7.5$ years, meaning that the disease progression model in Fig. 2 is shifted by $\pm 7.5$ years to account for the variability in the age of disease onset. The effects of the variance of the acceleration factors, and the two independent components of the space-shifts are illustrated in Fig. 4. The acceleration factors shows the variability in the pace of disease progression, which ranges between 7 times faster and 7 times slower than the average. The first independent component shows variability in the relative timing of the cognitive impairments: in one direction, memory and concentration are impaired nearly at the same time, followed by language and praxis; in the other direction, memory is followed by concentration and then language and praxis are nearly superimposed. The second independent component keeps almost fixed the timing of memory and concentration, and shows a great variability in the relative timing of praxis and language impairment. It shows that the ordering of the last two may be inverted in different individuals. Overall, these space-shift components show that the onset of cognitive impairment tends to occur by pairs: memory & concentration followed by language & praxis.

Individual estimates of the random effects are obtained from the simulation step of the last iteration of the algorithm and are plotted in Fig. 5. The figure shows that the estimated individual time-shifts correspond well to the age at which individuals were diagnosed with AD. This means that the value $p_0$ estimated by the model is a good threshold to determine diagnosis (a fact that has occurred by chance), and more importantly that the time-warp correctly register the dynamics of the individual trajectories so that the normalized age correspond to the same stage of disease progression across individuals. This fact is corroborated by Fig. 3 which shows that the normalized age of conversion to AD is picked at 77 years old with a small variance compared to the real distribution of age of conversion.

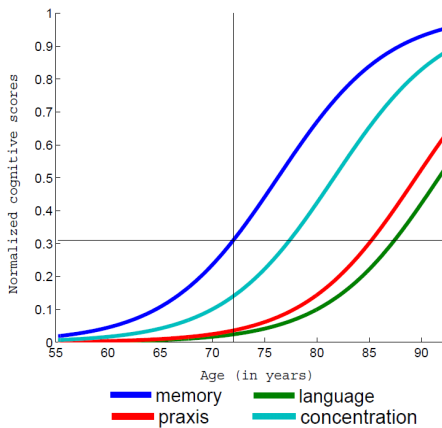

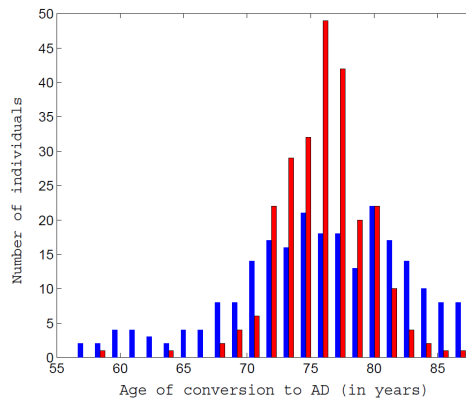

Figure 2: The four curves represent the estimated average trajectory. A vertical line is drawn at $t_0 = 72$ years old and an horizontal line is drawn at $p_0 = 0.3$.

Figure 3: In blue (resp. red) : histogram of the ages of conversion to AD ($t_i^{\text{diag}}$) (resp. normalized ages of conversion to $AD$ ($\psi_i(t_i^{\text{diag}})$)), with $\psi_i$ time-warp as in (1).

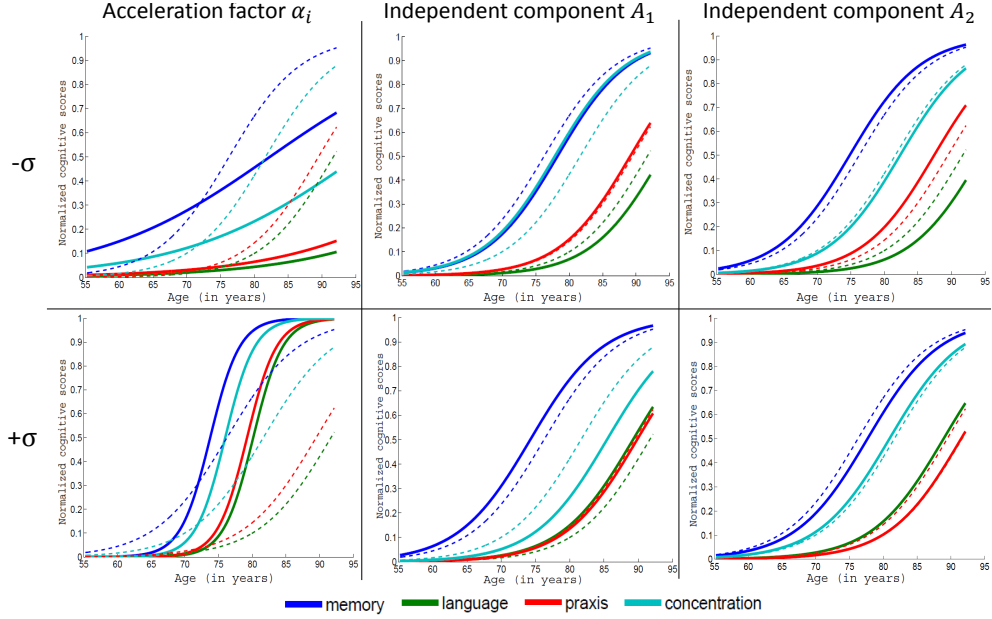

Figure 4: Variability in disease progression superimposed with the average trajectory $\gamma_\delta$ (dotted lines): effects of the acceleration factor with plots of $\gamma_\delta\big(\exp(\pm\sigma_\xi)(t-t_0)+t_0\big)$ (first column), first and second independent component of space-shift with plots of $\eta^{\pm\sigma_{s_i}\mathbf{c}_i(\mathbf{A})}(\gamma_\delta,\cdot)$ for $i=1$ or $2$ (second and third column respectively).

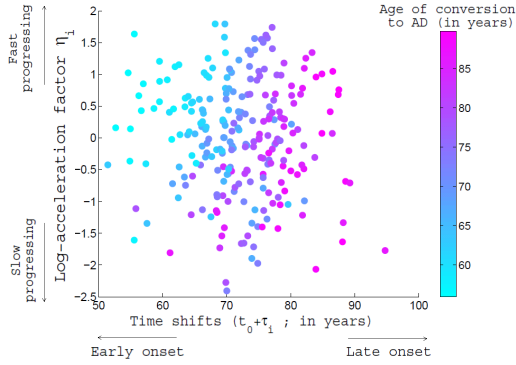

Figure 5: Plots of individual random effects: log-acceleration factor $\xi_i = \log(\alpha_i)$ against time-shifts $t_0 + \tau_i$. Color corresponds to the age of conversion to AD.

## 4.3 Discussion and perspectives

We proposed a generic spatiotemporal model to analyze longitudinal manifold-valued measurements. The fixed effects define a group-average trajectory, which is a geodesic on the data manifold. Random effects are subject-specific acceleration factor, time-shift and space-shift which provide insightful information about the variations in the direction of the individual trajectories and the relative pace at which they are followed.

This model was used to estimate a normative scenario of Alzheimer's disease progression from neuropsychological tests. We validated the estimates of the spatiotemporal registration between individual trajectories by the fact that they put into correspondence the same event on individual trajectories, namely the age at diagnosis. Alternatives to estimate model of disease progression include the event-based model [9], which estimates the ordering of categorical variables. Our model may be seen as a generalization of this model for continuous variables, which do not only estimate the ordering of the events but also the relative timing between them. Practical solutions to combine spatial and temporal sources of variations in longitudinal data are given in [7]. Our goal was here to propose theoretical and algorithmic foundations for the systematic treatment of such questions.

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
