[Reviews · NeurIPS 2015]

Submitted by Assigned_Reviewer_1

The authors propose a manifold model for longitudinal data analysis. The overall approach is a mixed-effects model; where each per-subject trajectory is constructed as a parallel transport of a "central" trajectory, with subject dependent temporal progression. While the model is quite complicated, the model parameters are interpretable. I found the proposed solution quite creative. I also found the paper well written. However, I am familiar with manifolds, and I fear the paper may be inaccessible for others in the community. I my opinion, the problem addressed is an important contribution to time series analysis. Unfortunately, I am unconvinced by the empirical evaluation.

- I would be quite interested in simulated data evaluation: to evaluate if the proposed approach correctly recovers model parameters corresponding to data sampled from a known manifold.

Such a simulated experiments would shed light on whether such structures are estimable from data under ideal conditions.

- I am unconvinced that the proposed approach is a good match for the presented application. While it seems this is not the focus of the paper, the authors might have been better served with a convincing simulated data experiment. How do the results of the presented approach compare to the results using a standard mixed effects model?

Minor concerns:

- I find it difficult to believe that Lemma 1 is a novel result. I suggest the authors clarify or provide an appropriate reference.

- Line 244: "to" => "to"

Suggestions:

- I am disappointed that the authors only evaluate a product manifold with a simple metric, although that might be a necessary tradeoff for parsimony and empirical feasibility. I would be interested in some guidance on how this approach may be extended to higher dimensional manifolds, perhaps as future work?

- A distribution on the Stiefel manifold e.g. the matrix Bingham-von Mises-Fisher distribution, might be a better choice for generating the "A" matrix than the presented ad-hoc procedure.
Summary: The authors propose a creative mixed-effects model for longitudinal data analysis. I like the technical presentation, but I am unconvinced by the empirical evaluation.

Submitted by Assigned_Reviewer_2

The paper proposes a class of mixed-effects models for longitudinal manifold-valued data, as well as a stochastic EM method for a special case. This is applied to real data on Alzheimer's disease progression. Spatiotemporal modelling is a very timely topic. It's a hot topic in medical imaging, and very relevant to longitudinal clinical datasets (as here).

Novelty: The methods here are related to those in [8] and [17], as the authors acknowledge. There are important differences, starting with the fact that the present paper applies to any Riemannian manifold, whereas [8] considers only diffeomorphism groups (appropriately, in an analysis of shape). More discussion of the relationship to previous work would be welcome. For example, lines 71-72 are puzzling: "Although the development of statistical models for manifold-valued data is a blooming topic (see [16], [17]), the construction of statistical models for longitudinal data on a manifold remains an open problem." (The title of [16] includes "longitudinal", as does [8].)

Also, you say of [8] in Line 75 "the variance of shapes does not depend on time whereas it should adapt with the average scenario of shape changes". I don't understand this statement, and how your scheme does allow time-dependent variance. (Do you mean \epsilon in Eqn (1)?).

The particular SAEM method seems somewhat novel to me, but the authors should discuss similarities and differences from [2].

I was disappointed that the geometric framework, while starting in a general manifold setting, moved to a product of 1-D manifolds, all with the same metric. I am left wondering whether the paper might have reached a wider audience if presented on R^N with less jargon. On a related point, is the multivariate logistic curves model really necessary? There exists a mapping from (0,1) to \R that transforms the metric in Section 2.4 to the Euclidean one, after which geodesics are straight lines, thus simplifying the exposition and, unless I've missed something, the SAEM algorithm (see line 245 and lines 251 onward).

Nonetheless, a general geometric framework can be very valuable. I like the idea of using parallel transport and exponentiation to parametrize the neighbourhood of a curve. There's a "tubular neighbourhood theorem" that says this is really a parametrization, at least if you stay close enough to the original curve; you should find and cite some version of it. The version I recall requires your vector w to be perpendicular to the curve, as you have required, though I presume more general versions exist. I am not convinced by your statements on lines 156-159. First: "The orthogonality condition ensures that a point on the parallel curve wi (, �) moves at the same pace as in the average trajectory." I don't think that's correct, if "pace" means speed. E.g. if \gamma is the equator of a sphere, then the parallel curves are latitudes, all parametrised by longitude, but with speeds proportional to their radii. The same-pace claim is of course true in a flat metric, as in the motivating example, but orthogonality is not required in that case. Second, on line 159: "vectors w with the same orthogonal projection in (t0) lead to the same geometric parallel curve but with a different time parameterization." I don't think that's correct either (e.g. on a sphere). Nonetheless, I think the desired conclusion, on lines 160-161: "spatial and temporal transformations commute" is correct, by definition.

The restriction of the typical disease progression to be a geodesic in \R^N, with the same metric on each dimension, seems quite restrictive. In the example, it means that (as the authors write in Section 2.3) "all biomarkers have on average the same dynamics but shifted in time" [and progressing more or less quickly]. That would seem an unreasonable assumption for many diseases. Nonetheless, it's a reasonable place to start, and has the great convenience of allowing a very low-dimensional parametrization of the mean dynamics. Note that the curve \gamma need not be a geodesic in the general geometric framework.

The adapted SAEM algorithm impresses me. Even if it's not actually needed for this example (as I suspect, see above), it seems valuable. Could the authors please clarify its similarities to and differences from citation [2], and comment on its wider applicability?

The experiment is interesting and topical. Fig 5 in particular suggests that the model is useful, demonstrating that the estimated time shifts correlate well with the age of conversion to AD, while the rate of progression doesn't.

The English is very good but not perfect. Some typos and minor comments are noted below.

p1 typo: "fourty"

p2, line 89: shall - > should?

p2, line 106: N-dim in R^N?? should be R^P for P > =N

pp2/3: a citation to some standard reference on Riemannian geometry would be helpful, one that includes the parallel curve concept in Definition 1.

p4, line 168: why choose a log-normal distribution? (no criticism, just a request for comment)

p4, line 173: why choose a Laplace distribution?

p4, line 181: could you choose another letter for \eta_i since \eta is used earlier for the parallel curve?

p4, line 181: remove subscript k from \epsilon

p5, line 231: where did A go?

p5, line 244: "prevents from resorting ot" - > "prevents us from resorting to"

p5, line 253: "shall write"? - > "may be written"?

p8, line 378: "p_0=" should be followed by "0.3"

Citations: capitalize Alzheimer's, EM algorithm, Riemannian ... and many others
Summary: A thoughtful contribution on an important topic. There are some flaws in the theory and discussion, and some of the content may not actually be needed for the given application, but the framework nonetheless seems valuable.

Submitted by Assigned_Reviewer_3

The authors propose a mixed-effects model for longitudinal progression on manifolds. their approach consists in modeling typical (average) trajectories in Riemannian space and being able to evaluate random effects (or variables) per individual which can describe the divergence of the individual trajectory from that populatiomn average. the variables learned for this puprose are acceleration variables, temporal offsets and spatial offsets. Inferring these variables allows the model to implicitly register patients with heterogeneous characteristics to common disease progression manifolds without enforcing a shared clock. Since the model is expensive and nonlinear, the authors use an established monte carlo EM scheme.

The authors use this method to model Alzheimer's data from the ADNI dataset, an established dataset of significance in neurological disease studies.

The paper is generally of high quality and well-researched. It is furthermore written clearly and is easy to read and understand.

The authors shows that their work is useful in the context of some experiments. However, it becomes apparent, that bayesian estimation of the involved variables would greatly benefit the model, since EM still is only maximizing parameters while a Bayesian approach could help identify multiple possible trajectories per patient.

In terms of novelty the paper is ambivalent: fundamentally all used techniques and the mixed effects model are well-known and no new technique is proposed. However, in the context of this application the model is useful and worth considering.

The paper is of middling significance to the NIPS community, it would probably attract more interest in a medically inspired venue.
Summary: The authors present a complex version of a mixed effects model and corresponding stochastic em estimation with an application in neurological disease progression modeling. The model succees in the stated goals and is valuable in the context of its application, but does not provide novel insights for ML.

Submitted by Assigned_Reviewer_4

This work is about estimating model parameters to data arising from Alzheimer's disease. The authors propose to assume that the data resides on a Riemannian manifold. The parameters of this manifold then characterize particular parameters, for instance the age when the disease occurs or the speed(s) with which brain functions (long/short-term memory, etc) deteriorate. Model parameters are estimated by a complicated probabilistic variant of the EM-algorithm.

In my opinion this paper is written somewhat confusingly. For instance, variables are re-used (\theta in lines 183 and 231). This complicates reading unnecessarily.

The authors could have more clearly indicated the type (scalar, vector, matrix) and the dimensions of their variables (I found, $A$, $\mathbf{A}_i$, and $\mathcal{A}_i$ particularly annoying).

Evaluation: Several variances are reported, corresponding to different variants of the algorithm. How can they be interpreted? I.e. is a value of \sigma=0.012 large or high? In my opinion, the authors should have more thoroughly evaluated whether the assumption of the Riemannian model really holds. For instance, error residuals would be very helpful. Another sanity check is to train with a part of the data and then check if the other data can be reasonably explained by the fitted model.

Fig. 3: What are normalized ages?
Summary: This work is about estimating model parameters to data arising from Alzheimer's disease.

The authors propose to assume that the data resides on a Riemannian manifold.

Submitted by Assigned_Reviewer_5

Strengths: An in depth and appropriate treatment of the problem at hand, with sufficient detail in the methodology. Overall this seems like a strong paper. Weaknesses: Given the small amount of data available, it seems like there are a lot of parameters to estimate, and a lot of modelling choices that are not immediately obvious. Without comparison to simpler methods, and analysis of the various modelling choices, it's hard to evaluate the experimental results.
Summary: This paper proposes to model the progression of Alzheimer's disease using a spatiotemporal mixed-effects model for manifold-valued data. The model is based on parallel curves on a Riemannen manifold, and inference is performed using a stochastic version of the EM algorithm.

Author Feedback
Author rebuttal: We warmly thank the reviewers for their detailed and insightful comments, which will serve to improve the presentation of the paper along the following lines.

R1 is concerned about (i) the relation with previous works, (ii) relevance of manifold structure for the application, (iii) possible technical flaws in the geometric framework, and (iv) originality and applicability of the MCMC-SAEM algorithm.

Ad (i): [8] does not respect the manifold structure of the diffeomorphism group, as the equivalent of our space-shift w is not transported along the manifold. Here, the variance of the space-shifts at any time-point is adjusted by the use of parallel transport. [17] estimates average trajectories but does not learn distribution of these trajectories in the spatiotemporal domain. [16] is not built on the inference of a statistical model.

Ad (ii): The logit function is the Riemannian logarithm taken at the inflexion point of the logistic curve. In our approach, this point is not fixed a priori, but is estimated by the algorithm as p_0. Even if we fix p_0 to be at the inflexion point, our model written in the Euclidean tangent-space is still *not* linear because of the multiplication of the acceleration factor and the time-shift. Therefore, the presentation in the Euclidean space would be more restrictive and would not greatly simplify the algorithm.

Ad (iii):
- We warmly thank the reviewer for pointing to us the "tubular neighbourhood theorem", which we were unaware of. It is relevant to our construction and the link will be clearly stated in the revision.
- "Pace" does not mean "speed" here, but the duration between two consecutive time-points, which would be the same on the average trajectory as between the related events on the subject-specific trajectories, regardless of the length of the curve between them.
- It would be more correct to say that the orthogonal projection of the vector w would play the same role as the acceleration factor, thus making the model not identifiable.

Therefore, we don't believe that there are flaws in the theory. Nonetheless, we will increase the clarity and precision of the presentation.

Ad (iv): The adapted MCMC-SAEM algorithm is essentially the same as in [2], which is designed for any mixed-effect generative model. Although presented here for data on a product manifold, its application for any other type of data and manifold is straightforward. This point will be better stressed in the revision.

R2 is concerned with the impact of the paper to the ML community. The usual mixed-effect models for longitudinal data are built on the idea of the regression of data against time, which is considered as a covariate. Here we propose a novel approach to learn distributions of trajectories in the spatiotemporal domain, where time is considered as a random effect. This approach may have important applications for the statistical analysis of spatiotemporal data, such as for the study of animal migration or diffusion of drugs in the body. A complete Bayesian framework would be very interesting by providing the parameter distributions, but is more demanding computationally and yields outputs that are less easily interpretable. The MAP estimator is an interesting tradeoff in such situations.

R3 is concerned with the notations, which will be edited according the reviewer's suggestions. We will report the percentage of variance explained to assess the ability of the model to explain the data. The fact that the model can predict the age at which patients are diagnosed with the disease shows its predictive power. This result obtained with 250 patients could be tested in a cross-validation setting, but would require another optimisation scheme to maximise the likelihood of a new data given the trained model.

R4 suggests simulated data experiments, which we did to check the validity of our framework. We will report them in the revision to better assess the identifiability of the model and the convergence of the algorithm.

R4, R5 and R6 are concerned with the comparison of the method w.r.t. simpler approaches. Even in the Euclidean case with a flat metric, our model essentially differs from usual longitudinal mixed-effect models, in that it remains non-linear and considers time as a random variable and not as covariate. Time warping methods do not estimate variation in measurement values together with time re-parameterization. The validity of the method is mostly assessed here by its ability to match the age at diagnosis of every patient near the same time-point in the average trajectory. The estimated parameters all have a clear interpretation and provide complementary insights into the data compared to existing methods.

We hope that the revision of the paper along the lines suggested by the reviewers will clarify the originality of the model compared to existing approaches, and better stress the applicability of the algorithm beyond the presented experiment.